# Traditional Diet and Environmental Contaminants in Coastal Chukotka I: Study Design and Dietary Patterns

**DOI:** 10.3390/ijerph16050702

**Published:** 2019-02-27

**Authors:** Alexey A. Dudarev, Sveta Yamin-Pasternak, Igor Pasternak, Valery S. Chupakhin

**Affiliations:** 1Department of Arctic Environmental Health, Northwest Public Health Research Center, St-Petersburg 191036, Russia; valeriy.chupakhin@gmail.com; 2Institute of Northern Engineering and Department of Anthropology, University of Alaska Fairbanks, Fairbanks, AK 99775, USA; syamin@alaska.edu; 3Institute of Northern Engineering and Department of Art, University of Alaska Fairbanks, Fairbanks, AK 99775, USA; gipasternak@alaska.edu

**Keywords:** subsistence food, traditional diet, indigenous people, cuisine, aesthetics, coastal Chukotka, Bering Strait, Russian Arctic

## Abstract

The article is the first in the series of four that present the results of a study on environmental contaminants in coastal Chukotka, conducted in the context of a multi-disciplinary investigation of indigenous foodways in the region of the Bering Strait. We provide an overview of the contemporary foodways in our study region and present the results of the survey on the consumption of locally harvested foods, carried out in 2016 in the Chukotkan communities of Enmelen, Nunligran, and Sireniki. The present results are evaluated in comparison to those of the analyses carried out in 2001–2002 in the village of Uelen, located further north. Where appropriate, we also draw comparative insight from the Alaskan side of the Bering Strait. The article sets the stage for the analyses of legacy persistent organic pollutants (POPs) and metals to which the residents become exposed through diet and other practices embedded in the local foodways, and for the discussion of the Recommended Food Daily Intake Limits (RFDILs) of the food that has been sampled and analyzed in the current study.

## 1. Introduction

### 1.1. Overview

Our multi-disciplinary team studies indigenous Yupik, Chukchi, and Inupiaq foodways in the region of the Bering Strait. We approach this subject from a number of perspectives, including aesthetics (culturally held ideas of beauty of an object or process), cultural identity, spirituality, social-ecological adaptation, food security, nutrition, and health. In the series of four articles, of which this is the first, we present the analysis of the data gathered in Chukotka, on the Russian side of the Bering Strait, and focus on the exposure to environmental pollutants among the indigenous residents through the consumption of locally harvested foods and water. Where appropriate, we also draw comparative insight from the Alaskan side of the Bering Strait. The current article provides an overview of the contemporary foodways in our study region and presents the results of the survey on the consumption of locally harvested foods, carried out in 2016 in the Chukotkan settlements of Enmelen, Nunligran, and Sireniki. With that, this article sets the stage for the analyses of legacy POPs and metals to which the residents become exposed through diet and other practices embedded in the local foodways [1,2] and for the discussion of the Recommended Food Daily Intake Limits (RFDILs) of the food that has been sampled and analyzed in the current study [3]. Together, articles I–IV, offer the perspectives from the several fields of sciences and humanities represented in our project.

### 1.2. Geography and Living Conditions in the Study Region

The locally harvested foods in the region of the Bering Strait rely on the ecologically similar coastal tundra environments and the shared marine resources of the northern Bering and Chukchi seas—the meeting point of the north Pacific and Arctic oceans. The Providensky district of the Chukotka Autonomous Okrug (Figure 1), where the 2016 sampling was carried out, occupies the southeast part of Chukotka Peninsula. Cape Chaplin is located 80 km west from the village of Gambell, sitting at the northwest tip of Saint Lawrence Island, Alaska. From the east the Providensky district is surrounded by Bering Sea, and from the south by Anadyrsky Bay. The length of the district coastline is 850 km, most of which is cut by rocky capes, deep bays and fjords. The total district territory is 26,800 km^2^. The number of inhabited localities is six; the largest town is Provideniya with population 2109 and the total district population number is 3752. The population density is 0.14 people/km^2^. Overall 56% of the population is urban and 44% rural; 58% are indigenous (mostly Chukchi and Yupik) and 42% are non-indigenous (mostly Russians) [4]. Eighty-two percent of the indigenous people live in rural areas. The demographic situation in Providensky district is characterized by continuous population decline due to outmigration (overwhelmingly by members of the non-indigenous population) and a decrease in the rates of natural population growth. The settlements where the current contaminant study was conducted are small: Enmelen and Nunligran are similar in population number (about 300 people total, more than 95% indigenous); Sireniki is slightly larger, and the percentage of indigenous people is 92%. The total population number in all three settlements decreased during 15 years (from 2000 to 2015) by 19–28%, while the number of indigenous population decreased by 13–26% (Table 1).

The territory of the district has deposits of gold, silver, tin, copper, arsenic, mercury, and uranium, but commercial mining is not carried out. Industry in the Providensky district is limited to production and distribution of energy and a handful of food items. The only roads in the district are the short gravel cover roads from Provideniya to the airport (12 km) and to Novoe Chaplino settlement (18 km), which have not been repaired for decades due to insufficient funding. Travel between Provideniya and other villages can be done with off-road vehicles or powerful trucks, and by helicopter. Over the years of our research, all of the settlements on the Chukchi Peninsula have been serviced by a single helicopter. The service is thus infrequent and the scheduled flights are delayed regularly due to maintenance and repair needs as well as weather conditions and subsequently the need to make a number of remedial flights before resuming the regular schedule. Ground travel is complicated by the overall scarcity of vehicles, their frequent breakage due to the rough conditions causing rapid wear and tear, difficulty in obtaining replacement parts, and shortage of fuel. Similar obstacles restrict boat travel, which is also subject to elaborate administrative regulations. In recent years, winter travel has become more precarious due to later freeze-up and thinner ice over the lakes and lagoons. Summer travel has also become more challenging due to the expansion of swampy and muddy conditions over the stretches of the routes affected by the permafrost thaw. 

There are no centralized electrical grids in the district; the settlements are powered by coal power stations which contaminate outdoor atmosphere, indoor air, and the snow and ice that during certain times become the source of water for general home use, including cooking and drinking. Many older and newer types of village homes are heated with individual coal stoves. Residents typically make a bulk purchase of their coal supply for the year (4 tons per residence) once it is brought it by barge. Storing large quantities of coal and hauling it under the normal windy conditions of the coastal Arctic is an everyday challenge. As mentioned earlier, wind-scattered coal dust is the chief reason residents say they need to walk some distance from the village to collect snow, when they need to make do in the absence of the regular water delivery. To some degree, especially while the stove is being filled, coal dust also spreads indoors. The coal stove location is typically in the kitchen area in the older homes and in a separate utility room in the newer cottages; in the latter case these rooms may also be the place of storage for the water barrels and various supplies, including food.

The municipal water delivery in the study settlements is facilitated through scheduled service by a water truck. Untreated water, pumped from nearby rivers and lakes, is typically unloaded into large barrels (200–250 L), kept inside a residence, using a hose. Water is removed with a ladle by dipping it into the surface of the water. Most residents use a wall hanging water dispenser (called *rukomojnik* in Russian), which they fill by scooping the water from the barrel with a ladle. The wastewater drains into a bucket, which is typically emptied outdoors on the ground. Sewage treatment facilities are absent in all of Chukotka—untreated sewage water flows into Anadyrsky Bay and the Bering Sea. In some villages the sewage drains go to cesspools, which periodically get pumped and discharged onto the ground surface or into the nearest body of water. The buckets used for the drainage and toilet are usually emptied near the habitation. Plastic bags to contain human waste in the “honey-buckets” are not used, and there are no special service for collecting and disposing of waste in communities. Some modern private cottages are equipped with toilets and septic systems [5].

### 1.3. Contaminants and Arctic Foodways, a Brief Summary

Indigenous people in the coastal communities of circumpolar regions are strongly dependent on the locally harvested foods, which provide nutrients and are important for maintenance of health, wellbeing, traditions, and cultural identity of the communities. However, many of the country foods, especially the blubber of marine mammals, are highly contaminated by persistent organic pollutants (POPs) and some metals. That is due to the bioaccumulation and biomagnification of POPs and metals in the marine food chain, as well as the lipophilic features of these contaminants. That is the reason for the high exposure to POPs and metals by native people who consume marine mammals [1,2].

In 2001–2002, community-based dietary and lifestyle surveys and an environmental exposure assessment that focused on local foods were conducted in coastal Chukotka, along with other Russian Arctic regions within the framework of the Russian Arctic persistent toxic substances (PTS) study [6]. The dietary survey (interviews of 251 indigenous people in the Uelen settlement) was based on self-reported daily (weekly, monthly) food frequencies. Nutritional patterns of the Yupik and Chukchi population residing in Uelen were for the first time presented at the Arctic Monitoring and Assessment Program (AMAP) conference “Impacts of POPs and mercury on Arctic environments and humans” in Tromso in January 2002 [7]. Subsequently, the main graph depicting the average annual consumption of local traditional foods by coastal Chukotka natives was reproduced in the AMAP Assessment Report 2009: Human Health in the Arctic [8] and in the monograph “Implications and Consequences of Anthropogenic Pollution in Polar Environments” [9]. The comprehensive article devoted to the dietary exposure to persistent organic pollutants and metals of Chukotka native people (based on the 2001–2002 collected data) was published in 2012 [10]. The Russian Arctic PTS study has revealed coastal Chukotka to have the highest (compared to other Russian Arctic regions) levels of contamination found in local foods and the highest contaminant exposure levels of local indigenous people. 

In 2016 in Enmelen, Nunligran, and Sireniki, 15 years after the data collection in Uelen, members of our team carried out a follow-up assessment of environmental legacy POPs and metals in the locally harvested sources of food, drinking water, indoor matter, and home-made alcoholic drinks. 

## 2. Materials and Methods 

### 2.1. Ethnographic Fieldwork

The ethnographic insight on the contemporary Bering Srtait foodways stems predominantly from the data gathered by co-authors Yamin-Pasternak and Pasternak in the course of our current collaborative project, launched in 2015. Where relevant, we also draw on the findings from the authors’ earlier research in the region, which spans two decades. Thus, the analyses of the questionnaire (discussed later in this article) and of the contaminants in food samples [1,2,3]—all collected in the Chukotkan settlements of Enmelen, Nunligran, and Sireniki—are considered in light of the harvesting and culinary practices, about which members of our team have learned from over 500 Yupik, Chukchi, and Inupiaq residents in the total of 18 communities on both Russian and Alaskan sides of the Bering Strait. The methods employed during Yamin-Pasternak’s ethnographic fieldwork include participant observation, semi-directed interviews, and the development of a public exhibition on the Bering Strait foodways, put together with our community-based collaborators [11]. 

### 2.2. Questionnaire

Community-based questionnaire aimed at collecting data on self-reported food intake frequencies has been completed by total of 112 persons: Enmelen 42, Nunligran 33, and Sireniki 37. Each participant was asked about the frequency of food intake of each item: 1–3 meals/day; 4–6 meals/week; 1–3 meals/week; 1–3 meals/month; 4–10 meals/year; and 1–3 meals/year. The average annual daily intakes for each foodstuff item were calculated using the self-reported intake frequencies, based on the hypothetically assumed portion size of 150 g/meal. Questionnaire also asked for detailed personal characteristics, including anthropometry, ethnicity, family, education, occupation, subsistence hunting and foraging and the associated methods used to preserve the locally harvested foods, lifestyle habits (such as domestic use of pesticides, smoking, alcohol drinking), risk perception of contamination of local food, water and environment, and self-assessment of health. Written informed consent was obtained from all participants of the study, which was performed in accordance with the Helsinki declaration. The study protocol was approved by the ethics review committee at the Northwest Public Health Research Centre (NWPHRC), St-Petersburg, Russia (ethical approval code 2016/01-2), and by the University of Alaska Fairbanks Institutional Review Board (ethical approval code UAF IRB # 645244).

The questionnaire-based interviews were conducted by co-authors Alexey A. Dudarev and Valery S. Chupakhin; both are health scientists with the NWPHRC. Russian language paper-printed questionnaires were used for survey.

Assistance in arranging the interviews in each of the settlements was provided by heads of administration in every village. The interviews were carried out using a standard scheme. First, a preliminary short explanation/presentation of the questionnaire by the NWPHRC representatives in a study group of respondents (local hunter collective, personnel of kindergarten, school, etc.) was given, with a clarification of nuances. Then, the paper questionnaires were distributed among the respondents who then independently (without conversing with others) filled in the questionnaires. The interviewers assisted each respondent needing clarification of a survey question or any step in the process. 

The majority of interviewees (Table 2) in all three settlements self-identified as Chukchi, which is an expected finding for the villages of Enmelen and Nunligran, as they were known throughout their history during the Soviet period to be predominantly Chukchi villages. The finding is somewhat unexpected for Sireniki, because this community is regarded the longest continuously occupied Yupik settlement [12] and continues in present times to be regarded as a Yupik village. The average age of the interviewed people was about 43 years (similar for men and women); the average duration of local residence was 35 years (38 for men and 33 for women). The average height of the respondents was 163 cm (170 for men and 158 for women); the average weight was 67 kg (69.3 kg for men and 65.3 kg for women). Overall, 44.6% of the respondents were married and 38.4% lived alone. Average number of family members in the studied group of three settlements was 4.1 people with a maximum 10 people; the average number of children was 2.1 children with maximum of eight children. The majority of families had two children. Secondary and vocational education prevailed among the respondents and nearly 20% of all interviewed persons had university education. Marine mammal hunters, school teachers, and kindergarten mentors were the three main occupations (60%) among the study cohort. In total, 62% of the respondents were engaged in family fishing and 54% in family hunting. Most of the interviewed people in collected mushrooms, berries, and wild plants in the summer, and many people preserved them for the winter time. 

## 3. Results

### 3.1. Dietary Pattern of the Studied Group

The contemporary cuisines in the Bering Strait utilize numerous species of marine and terrestrial mammals, fish, mollusks, ascidians, birds, fungi, seaweeds, and tundra plants. Subsistence activities take place year-round [13], with the period from of July through October usually being the busiest due to the abundance of the desired berries, greens, roots, and mushrooms that are in their prime for harvesting. The primetime for harvesting each of the targeted foods can be a period as short as a few days, and it may overlap with the prime harvesting time for other desired products. Procuring adequate quantities of certain berries, mushrooms, and greens requires hours of foraging across vast territories of the tundra, which most residents do on foot. Many of the products must be processed immediately after the harvest to prevent spoilage. Even at their peak, the collecting activities must take place alongside fishing, hunting, and the work of processing various kinds of catch, maintenance and repair of the tools and infrastructure (such as drying racks and smokehouses), and the everyday responsibilities of childrearing, eldercare, and household chores. Because of all of those circumstances, during the peak seasons for the gathering of fungi and plants people often work around the clock. The intensity of the peak season may make other times of year feel like downtime, although people hunt, catch fish, and collect desired seaweeds and ascidians throughout the year. During any season, the harvesting and processing of most products require laborious effort and highly specialized knowledge and set of skills. Foods are preserved by air-drying, salting and pickling, smoking, freezing, and by various methods of fermentation, with many recipes involving two or more of the preservation means (for example, fish can first be allowed to age, then pickled in brine, and then hung in the smokehouse for the final curing). Preserving food safely while also achieving the desired gastronomical and aesthetic appeal entails an intricate understanding of how the local climatic conditions interact with the physical properties of food, built environment, storage vessels, and other tools of material culture. 

Everyday meals in most instances are consumed with members of the family and guests gathering around a communal platter, which can be a carved wooden dish, a cutting board, enamel or aluminum bowl, or a standard manufactured serving tray. Most or all of the meal’s ingredients are laid out on a platter (or multiple bowls and platters, in the case of a larger feast), cut into bite-size pieces, with the expectation for every eater to reach and grab, at their own pace, using their fingers (Figure 2, Figure 3 and Figure 4). A single meal may feature several types of marine mammals (prepared by different means: aged, aged and then boiled, boiled after having been harvested and/or frozen, dried), reindeer and other terrestrial mammals and birds (aged and/or boiled or roasted), fish (dried, aged, smoked, frozen-sliced, or boiled), plant products (fresh or frozen greens, roots, combinations of several tundra plants mashed with blubber, blood, and fish raw), and a category of foods colloquially referred to as “gifts of sea” or just “seafood.” The last includes various types of crustaceans, including the mollusks taken from walrus stomach, and tide-carried marine flora and fauna, harvested predominantly along the seashore following a storm. Kelp (which may be procured while boating near the shore or collected on the beach) is appreciated for the flavor it adds to the meat and fish broths; boiled seaweeds are featured on meat platters and eaten in combination with other foods (Figure 3). Most ascidians are eaten freshly collected, shortly following the harvest; they are considered a treat (Figure 5). In the instances where they are frozen for future consumption, the leftovers are eaten frozen-uncooked, as part of a larger meal to which they are said to add the desired saltiness, several kinds of unique textures, and the aroma of sea. The ascidian of special cultural importance is *Halocynthia aurantium*, known locally as *upa* (Figure 6). Harvesting *upa* is an activity in its own rite, which involves long outings on the sea ice, repeatedly lowering a rake attached to a long rope in the hole in the ice in an attempt to scrape a cluster of these bright orange tunicates from the sea bottom. The village of Novoe Chaplino (also part of the Providensky district though not surveyed in the current study) is especially famous for its abundance of *upa* [14] and in 2017 was the site of *Upa*fest—a regional festival dedicated to the associated activities and dishes (an array of salad-like combinations with both local and store-bought ingredients, variably dressed) associated with this product. Residents of Novoe Chaplino provide *upa* to their relatives and friends in other villages, including the three that are surveyed here. The chart in Figure 7 shows the consumed amounts of foods in each category, reported by the 2016 survey respondents in Enmelen, Nunligran, and Sireniki.

### 3.2. Quantitative Assessment of Local Food Consumption

The average total annual amount of local foods consumed by a native resident at our three study sites is 191.5 kg. Fish and marine mammals constitute the highest percentage of the consumed local food (62%); other six food groups account for 10 to 15%. Compared to the 15-year old data on the residents of Uelen (located right on the Bering strait 250 km to the north), who in 2001 consumed 226.4 kg/person/year of local foods [10], the native people of Providensky district consume 30 kg less of the locally harvested foods, with the consumption of marine mammals being twice less than in the 2001 sample. There are several possible reasons for this: the villages in the north of Chukotka are generally known for their highly skilled hunting crews and the strong administrative leadership, and are praised for their ability to secure good hunting equipment and harvest quotas; 2001 was a year closely following the severe socio-economic crisis that erupted after the collapse of the Soviet Union, when the locally harvested foods were often the only source of nutrition in Chukotkan villages; and subsistence harvests vary from year to year, affecting the percentages of local and store-bought food (when available) in the overall diet.

Today, all indigenous residents of Chukotka eat a mixed diet of indigenous and Slavic/Eastern European foods, as well as a limited variety of mass-manufactured convenience foods. The village stores are serviced largely with the bulk yearly deliveries by barge. Occasionally all-terrain vehicle shuttles make steeply cost-prohibitive supply runs from the regional hubs. It is common for the frozen and canned goods sold at village stores to show a past due expiration date in the range between several months to several years. The area where the Chukotkan residents believe they have an overwhelming advantage is the presence of a municipally managed bakery in every village. These bakeries, which have been in continuous operation since Soviet times, produce crispy sourdough and rye bread on daily basis—food that is considered essential and that has been in high demand since the introduction of flour by the Russian–American traders in the region in the second half of the 19th century [15]. By comparison, in the villages on the Alaskan side of the Bering Strait, the only store-bought options for bread products are mass-manufactured soft bread and “pilot crackers.” The village stores offer small, gastronomically restrictive inventories, especially in fresh produce, while representing some of the most expensive retail in the world. Most families use their village stores for periodic bulk purchases of the main staples, such as flour, other grains, powdered milk and eggs, dehydrated potato, cooking oil, and tea and coffee. From day to day, village residents may pick up items such as packaged pasta, instant noodle soup, butter-like spread, low quality sausage and cheese products, frozen meat (chicken, beef, or pork, often showing discoloration and other signs of having been “freezer-burned”), and nominally fresh produce items. For many households, it is common to be cash-poor for a period of time from days to weeks when they are not able to buy anything from the store. 

## 4. Discussion

The core strength of broaching the subject of our study with the diverse disciplinary expertise represented on our team is that we are able to offer a spectrum of qualitative and quantitative insights, along with their interpretive contexts, stemming from several fields of social and natural sciences and humanities. Hence, the consumed amounts of the different kinds of locally harvested foods, reported by the survey participants, can be regarded as part of a larger and complex picture of the dietary practices, every-day and festive menus, structure of mealtimes, eating customs, taste preferences, social and spiritual considerations, and the many ways in which people create beauty in life through the work of harvesting, processing, and preparing local foods. The culinary principles and broader aesthetics of serving and eating local food vary between households and communities, but also share certain established forms. Understanding this patterns will become especially important once we consider the parameters for recommending the intakes of certain foods in light of our findings on the presence of environmental contaminants in the food sources of the indigenous people of Chukotka [3]. 

The overall extent to which individuals and families mix store-bought and local foods in their diet varies to a high degree and is determined by a complex array of intersecting factors, including age, occupation, monetary income, time of year, access, experience with state institutions such as boarding school, family relationships, personal preferences, and culturally held ideas about food, health, and status in society. Attitudes toward certain foods vary among individuals, members of different generations, and people of different occupations [16]. They are also based on a number of other parameters including the concern for contaminants. Nevertheless, in every community where we have worked, and as has been stated by researchers working in other regions of the Arctic [17,18,19,20,21], people share a widely held belief that eating locally harvested foods is a core cultural expression, important for nutrition, spirituality, social ties, quality of life, and survival in a physical sense and as a people. It is not only eating, but the entire way of life within which local foods and cuisines are embedded—being out on the tundra and at sea, advancing and passing on the knowledge of environmental conditions and the skills of hunting and foraging, and the associated labor and cooperation necessary to maintain tools and equipment and to procure and process foods, as well as all the inextricable social relationships—is considered healthful and essential. Most importantly, the spiritual grounding of the indigenous foodways is part of the indigenous cosmology, where humans and animals are tied by the circulation of souls and kinship bonds, and the souls of the late relatives live in the animals who, when they are hunted, return to their families to care and provide for them [22]. 

Alongside the centrality of the locally harvested foods, the factors to consider include the poor quality of the foods sold at the village store, and the prohibitively high cost and limited availability of the foods offered for purchase. The remoteness of the region from the perspective of the commercial supply routes and the scarce transportation options regularly cause lapses in deliveries. Weeks and even months may go by before a depleted supply of meat and produce is replenished, restricting the store inventory to basic non-perishables, such as grains and canned goods showing old expiration dates. Together, these factors amount to a scarce set of alternatives for the individuals interested in pursuing the dietary options outside the realm of the locally harvested foods. 

## 5. Conclusions

The qualitative and quantitative data presented aim at outlining the dietary practices at our study sites in the context of a broader portrayal of the contemporary Bering Strait cuisine. The descriptions of the geographic setting, history of the Soviet and post-Soviet developments, recent demographic shifts, utilities and transportation infrastructure, and current living conditions should convey a sense of the challenges that residents in our study region confront in the course of everyday living. The current article is intended to set the stage for the three other articles included in this volume that discuss the legacy POPs [1] and metals [2] found in the food samples obtained at our study sites, and the recommendations on the daily intake of foods and other steps to address the threat posed by the presence of environmental contaminants in the Chukotka native diet [3]. It is important that the data and interpretive ideas presented in the subsequent article are considered in light of the circumstances described here. Most critical is that in moving ahead with examining our study results, the reader never loses sight of the fact of the integral role that the locally harvested foods play in the wellbeing, identity, and way of life of the native people of Chukotka.

## Figures and Tables

**Figure 1 ijerph-16-00702-f001:**
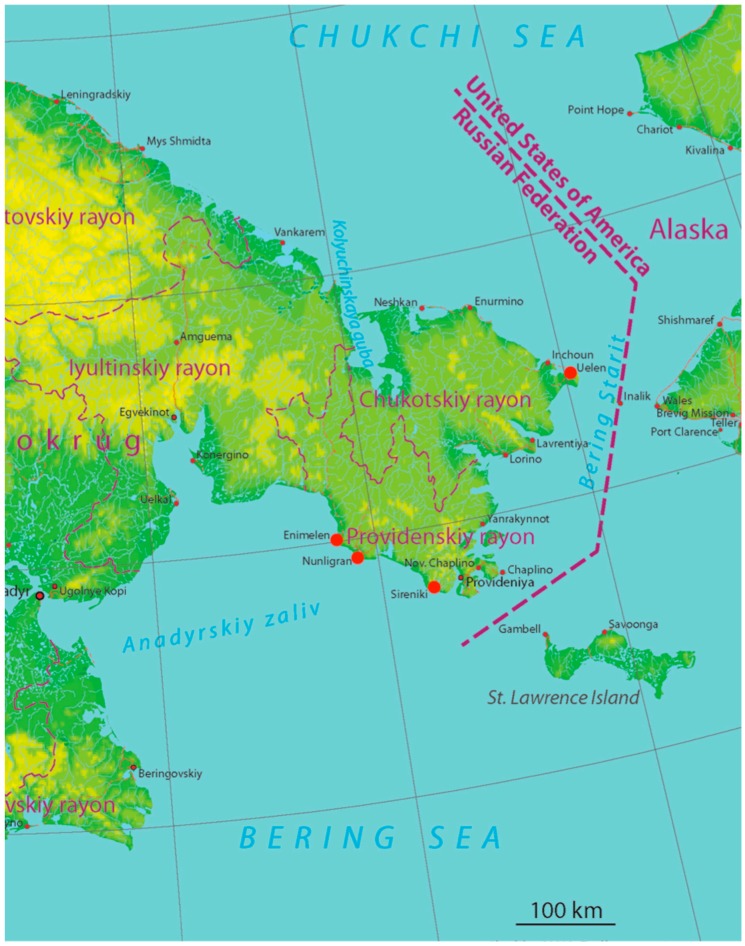
Study settlements (red circles) in Providensky district (2016) and Chukotskiy district (2001–2002) of Chukotka okrug.

**Figure 2 ijerph-16-00702-f002:**
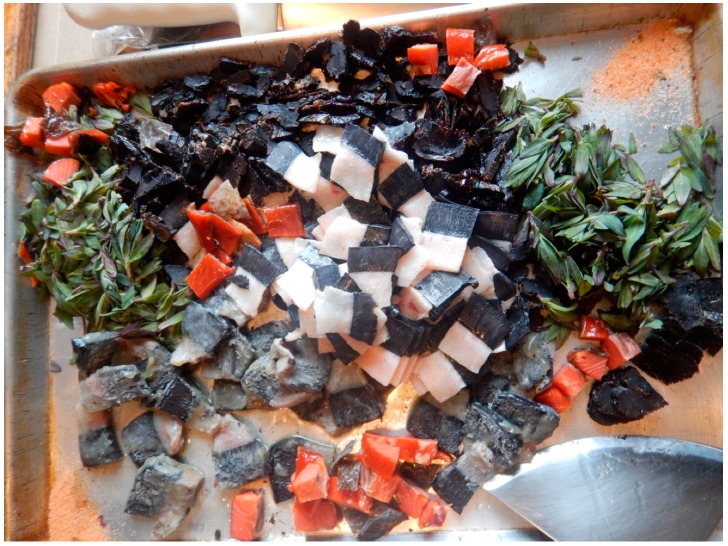
A meal tray/platter featuring raw-frozen mantak (whale skin with a thin layer of adjacent blubber), aged mantak, salmon, dried meat of a bearded seal, fireweed leaves (Chamerion latifolium), and store-bought seasoned salt in the upper right corner. Photo by Igor Pasternak and Sveta Yamin-Pasternak, 2016.

**Figure 3 ijerph-16-00702-f003:**
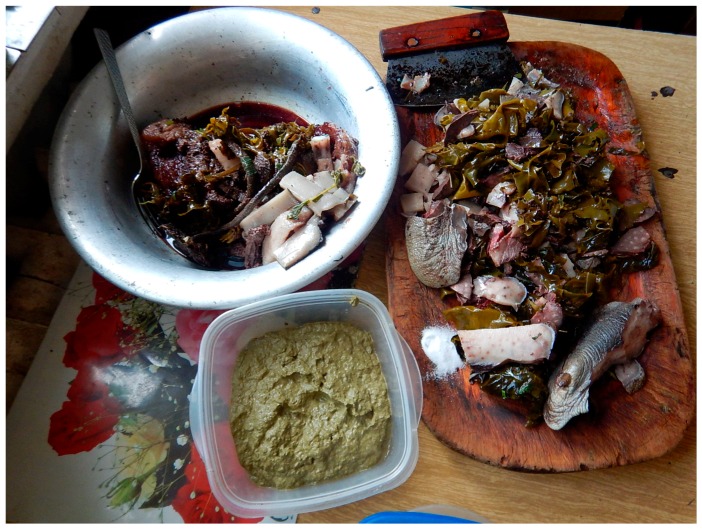
A meal featuring several parts of walrus, including sections of a flipper, fibrous tissue located between skin and blubber, meat, and kidney—all parboiled and served with green kasha (see Section 3.2). Photo by Igor Pasternak and Sveta Yamin-Pasternak, 2015.

**Figure 4 ijerph-16-00702-f004:**
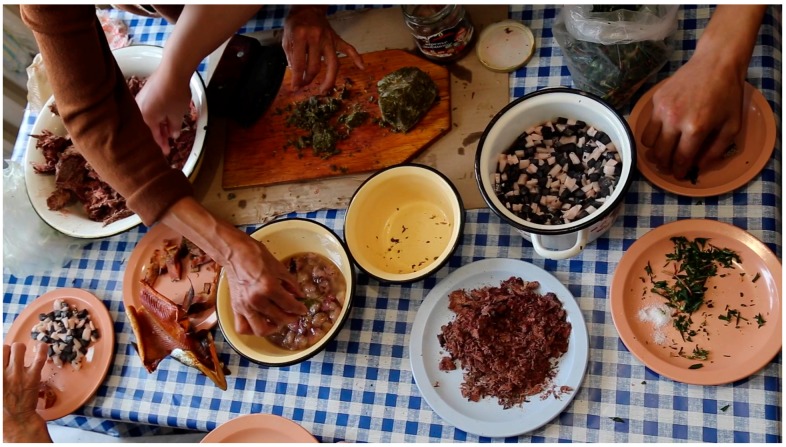
A meal featuring smoked salmon, a mix of dried walrus and whale meat, mantak, walrus blubber pieces in oil (rendered from walrus blubber), fresh-frozen fireweed leaves, and chopped leaves of boiled-frozen sourdock (*Rumex arcticus*), eaten with small amounts of plain store-bought salt, to taste. Photo by Igor Pasternak and Sveta Yamin-Pasternak, 2015.

**Figure 5 ijerph-16-00702-f005:**
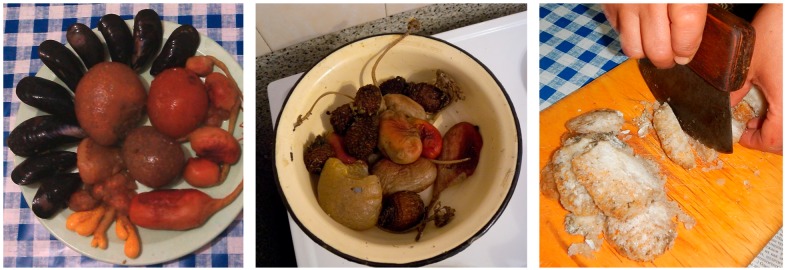
Assorted “Gifts of Sea”: mussels and ascidians. Photo by Igor Pasternak and Sveta Yamin-Pasternak, 2015.

**Figure 6 ijerph-16-00702-f006:**
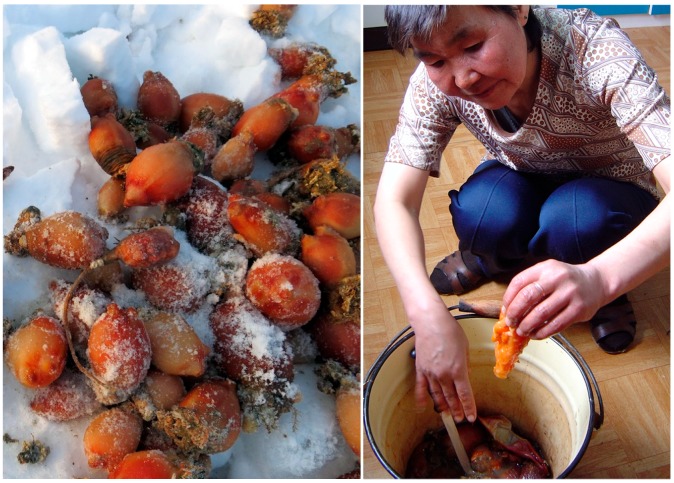
*Halocynthia aurantium*, known locally as *upa*, freshly caught from under the sea ice (left) and being pealed for immediate consumption (right). Photo by Sveta Yamin-Pasternak, 2004.

**Figure 7 ijerph-16-00702-f007:**
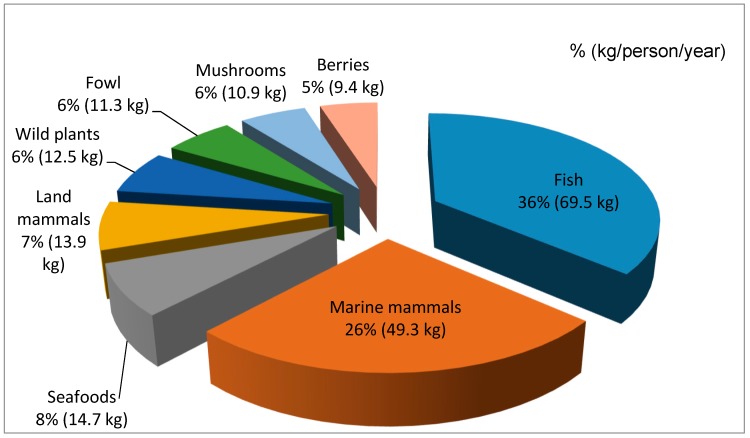
Structure (%) and average annual consumption of local foods (kg/person/year) by coastal native people residing in the settlements of Enmelen, Nunligran, and Sireniki.

**Table 1 ijerph-16-00702-t001:** Dynamics of the population number of the studied settlements Enmelen, Nunligran, and Sireniki [4].

Population	Years	Enmelen	Nunligran	Sireniki
Total Population	2000	374	371	526
	2005	341	311	458
	2010	328	288	391
	2015	297	300	382
Indigenous Population	2000	346	338	476
	2005	317	294	432
	2010	312	281	350
	2015	283	294	352

**Table 2 ijerph-16-00702-t002:** Characteristics of the respondents in three settlements (Enmelen, Nunligran, and Sireniki).

Characteristics	Attribute	Value	Enmelen (*n* = 42)			Nunligran (*n* = 33)			Sireniki (*n* = 37)			Totally 3 Settlements (*n* = 112)		
			*n* or Mean	% or SD *	Min–Max	*n* or Mean	% or SD *	Min–Max	*n* or Mean	% or SD *	Min–Max	*n* or Mean	% or SD *	Min–Max
Ethnicity	Chukchi	*n* (%)	36	85.7%		25	75.8%		15	40.5%		76	67.9%	
	Eskimo	*n* (%)	1	2.4%		1	3.0%		14	37.8		16	14.3%	
	Russians	*n* (%)	0	0		5	15.2%		3	8.1%		8	7.1%	
	others	*n* (%)	5	11.9%		2	6.1%		5	13.5%		12	10.7%	
Gender and age (years)	Men	*n* (%)	20	47.6%		11	46.4%		18	48.6%		49	43.8%	
		age	44.8	9.1 *	26–57	45.6	10.9 *	28–58	40.2	10.1 *	18–56	43.3	10.0 *	18–58
	Women	*n* (%)	22	52.4%		22	53.6%		19	51.4%		63	56.2%	
		age	45.8	11.2 *	27–72	39.3	12.7 *	21–59	41.7	11.3 *	23–59	42.3	11.9 *	21–72
	M + W	*n* (%)	42	100%		33	100%		37	100%		112	100%	
		age	45.3	10.2 *	26–72	41	12.3 *	21–59	40.9	10.6 *	18–59	42.7	11.1 *	18–72
Duration of local residence	Men	years	40.7	12.2 *	16–57	40.5	11.4 *	21–58	33.3	13.9 *	4–49	38.1	12.9 *	4–58
	Women	years	39.6	18.5 *	1–72	28.8	17.9 *	0–61	29	17.3 *	1–59	32.6	18.4 *	0–72
	M + W	years	40.1	15.7 *	1–72	32.9	16.7 *	0–61	40	15.8 *	1–59	35	16.4 *	0–72
Height, cm	Men	cm	168.3	6.5 *	158–180	171.1	2.1 *	167–174	170.5	7.5 *	162–188	169.7	6.2 *	158–188
	Women	cm	156.7	6.8 *	140–170	158.3	6.3 *	148–168	159.1	6.6 *	146–172	157.9	6.6 *	140–172
	M + W	cm	162.3	8.8 *	140–180	162.4	8.1 *	148–174	164.6	9.0 *	146–188	163.1	8.7 *	140–188
Weight, kg	Men	kg	65,4	9.0 *	53–90	72.8	16.9 *	58–110	71.7	15.4 *	45–106	69.3	13.5 *	45–110
	Women	kg	66.9	11.5 *	45–87	62.8	14.5 *	43–100	65.9	15.8 *	45–100	65.3	13.8 *	43–100
	M + W	kg	66.2	10.3 *	45–90	66.1	15.7 *	43–110	68.6	15.6 *	45–106	67	13.7 *	43–110
Marital status	married	*n* (%)	16	38.1%		10	30.3%		24	64.9%		50	44.6%	
	cohabiting	*n* (%)	8	19.0%		5	15.2%		2	5.4%		15	13.4%	
	single	*n* (%)	16	38.1%		17	51.5%		10	27.0%		43	38.4%	
Family members	average	*n* (SD *)	4.3	1.8 *	1–10	3.6	1.6 *	1–8	4.3	1.7 *	1–7	4.1	1.7 *	1–10
Children in family	1	*n* (%)	11	26.2%		10	30.3%		4	10.8		25	22.3	
	2	*n* (%)	13	31.0%		10	30.3%		8	21.6		31	27.7	
	3	*n* (%)	5	11.9%		3	9.1%		11	29.7		19	17.0	
	4	*n* (%)	4	9.5%		1	3.0%		1	2.7		6	5.4	
	5 and >	*n* (%)	2	4.8%		1	3.0%		2	5.4		5	4.5	
	average	*n* (SD *)	2.4	1.6 *	0–8	1.8	1.3 *	0–6	2.2	1.3 *	0–5	2.1	1.5 *	0–8
Education	no	*n* (%)	1	2.4%		0	0		1	2.7%		2	1.8%	
	primary	*n* (%)	1	2.4%		1	3.0%		0	0		2	1.8%	
	secondary	*n* (%)	17	40.5%		12	36.4%		9	24.3%		38	33.9%	
	vocational	*n* (%)	10	23.8%		8	24.2%		13	35.1%		31	27.7%	
	university	*n* (%)	7	16.7%		8	24.2%		7	18.9%		22	19.6%	
Occupation	marine mammal hunters	*n* (%)	15	35.7%		9	27.3%		5	13.5%		29	25.9%	
	school teachers	*n* (%)	8	19%		9	27.3%		6	16.2%		23	20.5%	
	kindergarten mentors	*n* (%)	5	11.9%		6	18.2%		4	10.8%		15	13.4%	
	technicians, mechanics, machinists	*n* (%)	3	7.1%		0	0		9	24.3%		12	10.7%	
	clerks, secretaries	*n* (%)	3	7.1%		3	9.1%		2	5.4%		8	7.1%	
	sellers, marketers, shop workers, cooks	*n* (%)	2	4.8%		3	9.1%		2	5.4%		7	6.3%	
	janitors, cleaners, guards	*n* (%)	1	2.4%		1	3%		3	8.1%		5	4.5%	
	others	*n* (%)	3	7.1%		0	0		1	2.7%		4	3.6%	
Family fishing	weekly	*n* (%)	16	38.1%		5	15.2%		0	0		21	18.8%	
	monthly	*n* (%)	4	9.5%		2	6.1%		6	16.2%		12	10.7%	
	4–8 times/year	*n* (%)	11	26.2%		7	21.2%		12	32.4%		30	26.8%	
	1–3 times/year	*n* (%)	5	11.9%		7	21.2%		9	24.3%		21	18.8%	
	never	*n* (%)	6	14.3%		11	33.3%		14	37.8%		31	27.7%	
Family hunting	weekly	*n* (%)	11	26.2%		7	21.2%		10	27.0%		28	25.0%	
	monthly	*n* (%)	7	16.7%		4	12.1%		2	5.4%		13	11.6%	
	4–8 times/year	*n* (%)	1	2.4%		3	9.1%		3	8.1%		7	6.3%	
	1–3 times/year	*n* (%)	9	21.4%		2	6.1%		2	5.4%		13	11.6%	
	never	*n* (%)	14	33.3%		17	51.5%		20	54.1%		51	45.5%	

Note: gray background color was used for better visual segregation of the data characterizing the separate settlement. *n*: number; mean: arithmetic mean; * standard deviation (SD).

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
