# Peer review of "Traditional Diet and Environmental Contaminants in Coastal Chukotka I: Study Design and Dietary Patterns"

_ijerph, 2019, doi:10.3390/ijerph16050702_

Round 1

Reviewer 1 Report

Ethnographic Fieldwork, Questionnaire, Dietary pattern of the studied group and Quantitative assessment of local foods consumption were reviewed in this paper. The food samples and the recommendations on the daily intake of foods provides the basis for environmental contaminants in the Chukotka Native diet. The paper is innovative and continuous. Comment is as follows:

1.It is suggested that the passive voice should be used more frequently in the paper. e.g., “The article is the first in the series of four that present the results of a study on 12 environmental contaminants in coastal Chukotka”(line 12-13);“We provide an overview of  the cotemporary foodways in our study region and present the results of the survey on the consumption of locally harvested foods,” (line 14-16); 2. It is suggested that the present tense should be replaced by the past tense. e.g.,“The present results are evaluated in comparison to those of the analyses carried out in 2001-2002 in the village of Uelen, located further north” (line 17-18);” Our multi-disciplinary team studies Indigenous Yupik, Chukchi, and Inupiaq foodways in the region of the Bering Strait.” (line 27-28);”

3. It is suggested that the respondents should be divided into children group and adults group in Table 2. After all, exposure risk for children and adults might be different.

Author Response

Review Report 1

Comments and Suggestions for Authors

Ethnographic FieldworkQuestionnaireDietary pattern of the studied group and Quantitative assessment of local foods consumption were reviewed in this paper. The food samples and the recommendations on the daily intake of foods provide the basis for environmental contaminants in the Chukotka Native diet. The paper is innovative and continuous.

Comment is as follows: 1.It is suggested that the passive voice should be used more frequently in the paper. e.g., “The article is the first in the series of four that present the results of a study on 12 environmental contaminants in coastal Chukotka”(line 12-13); “We provide an overview of the cotemporary foodways in our study region and present the results of the survey on the consumption of locally harvested foods,” (line 14-16);

Answer: We believe this suggestions stems from a particular writing convention, different from the one the authors have learned in the course of our substantial training in academic and scientific writing. The latter actually recommends the opposite: to avoid passive voice as much as possible. We believe that both conventions are valid and there is not one that is more or less appropriate than the other for being implemented here. We leave it up to the journal editor to make the final decision in this regard.  

2. It is suggested that the present tense should be replaced by the past tense. e.g.,

“The present results are evaluated in comparison to those of the analyses carried out in 2001-2002 in the village of Uelen, located further north” (line 17-18);” Our multi-disciplinary team studies Indigenous Yupik, Chukchi, and Inupiaq foodways in the region of the Bering Strait.” (line 27-28);”

Answer: Different writing conventions suggest various combinations of tense usage in scientific writing. We adhere to the practice of using simple present tense for stating study objectives, discussing known literature, and relating the circumstances that are true at the time of writing and are likely to continue to be true beyond the time of writing (such as the fact that our team studies food ways in the region of the Bering Strait). It is one of the existing, widespread, established practices of tense usage in academic and scientific writing.

3. It is suggested that the respondents should be divided into children group and adults group in Table 2. After all, exposure risk for children and adults might be different.

Answer: The reviewer appears to have overlooked the fact that all respondents listed in Table 2 are adults, ages 26-72, while the section “children in family” provides lists the number and percentage of children in the respondent’s family.

Gender and age   (years)

Men

n (%)

20

47,6%

age

44,8

9,1*

26-57

Women

n (%)

22

52,4%

age

45,8

11,2*

27-72

M+W

n (%)

42

100%

age

45,3

10,2*

26-72

Reviewer 2 Report

In this paper, authors described the social characteristics of 3 human population living in coastal Chukotka.

The introduction is not a classical "state of the art", but a description of the study. I think most of it should be moved within the section "Methods". Since the objectives of the four studies are lines 27 to 41, all the following could be moved.

Also, lines 112-114, authors write about the contamination of marine mammals by pollutants (POPs and metals). References are necessary. I'm sure authors can easily find tens of studies conducted in Arctic regions.

Finally, lines 117-130, authors describes the references, while reader needs information about what is written within those references. For instance, authors wrote: "Subsequently, the main graph depicting the average annual consumption of local traditional  foods by Coastal Chukotka Natives was reproduced in the AMAP Assessment Report 2009: Human Health in the Artctic [5] and in the monograph 'Implication and consequences of Anthropogenic Pollution in Polar Environment [6]' ". I think that notations [5] and [6] provides the references, the text should provide the main information from those reports.

Maybe because I have no knowledge about ethnography, I am a skeptical about the scientific approach in the sentence line 335-339: "Most importantly, the spiritual grounding of the Indigenous food ways is part of the Indigenous cosmology, where humans and animals are tied by the circulation of souls and kindship bonds, and the souls of the late relatives live in animals who, when they are hunted, return to their families to care and provide for them [19].

Author Response

Review Report 2

Comments and Suggestions for Authors

In this paper, authors described the social characteristics of 3 human population living in coastal Chukotka.

·         The introduction is not a classical "state of the art", but a description of the study. I think most of it should be moved within the section "Methods". Since the objectives of the four studies are lines 27 to 41, all the following could be moved.

Answer: We believe that all the information presented in the Introduction section properly belongs in this section because it introduces the study, provides a brief description of the study region, and helps situate the present study within the scope of the historically and geographically focused questions/problems the present study is trying to address. Instead of moving parts of the introduction into the Methods section, as suggested by the reviewer, we divided the Introduction into sub-section to represent the several introductory topics presented.

·         Also, lines 112-114, authors write about the contamination of marine mammals by pollutants (POPs and metals). References are necessary. I'm sure authors can easily find tens of studies conducted in Arctic regions.

Answer: Articles II and III submitted for the same issue of the journal provide extensive bibliographic references for the research on the corresponding topics. The Abstract and Introduction in the current article clearly states that this is the first in the series of four and makes several references to the Articles II, III, and IV in “this issue.” We believe the reviewer is overlooking this important context.

·         Finally, lines 117-130, authors describes the references, while reader needs information about what is written within those references. For instance, authors wrote: "Subsequently, the main graph depicting the average annual consumption of local traditional  foods by Coastal Chukotka Natives was reproduced in the AMAP Assessment Report 2009: Human Health in the Artctic [5] and in the monograph 'Implication and consequences of Anthropogenic Pollution in Polar Environment [6]' ". I think that notations [5] and [6] provides the references, the text should provide the main information from those reports.

Answer: We disagree with the comment, including the reviewer suggestion “the text should provide the main information from these reports” represents a misunderstanding on behalf of the reviewer. The diagram to which we are referring represents is a diagram so well-known internationally within the science and policy discussions on the Arctic that it has been reproduced in multiple important publications and international documents, which are precisely the publications and documents we are listing within the said manuscript lines. And the description of the diagram itself is provided following Figure 7, within the lines 274-286.

·         Maybe because I have no knowledge about ethnography, I am a skeptical about the scientific approach in the sentence line 335-339: "Most importantly, the spiritual grounding of the Indigenous food ways is part of the Indigenous cosmology, where humans and animals are tied by the circulation of souls and kindship bonds, and the souls of the late relatives live in animals who, when they are hunted, return to their families to care and provide for them [19].

Answer: We believe the reviewer correctly attributes the expressed skepticism to the reviewer’s lack of familiarity with ethnographic research in the Arctic and with the cultures and belief systems of the Indigenous peoples in our study region. The manuscript co-authors Yamin-Pasternak and Pasternak have over twenty years of experience in conducting ethnographic research on this topic, and are therefore able to authoritatively put forward the said statement, which also aligns with a widely documented perspective in Indigenous and anthropological scholarship.